# Optimal Design of Multiple Floating Rings for 3D Large-Area Trench Electrode Silicon Detector

**DOI:** 10.3390/s22176352

**Published:** 2022-08-24

**Authors:** Wenzheng Cheng, Manwen Liu, Zheng Li, Zhenyang Zhao, Zhihua Li

**Affiliations:** 1Institute of Microelectronics, Chinese Academy of Sciences, Beijing 100029, China; 2School of Integrated Circuits, University of Chinese Academy of Sciences, Beijing 100049, China; 3School of Physics and Optoelectronic Engineering, Ludong University, Yantai 264025, China; 4School for Optoelectronic Engineering, Zaozhuang University, Zaozhuang 277160, China; 5Shandong Dongyi Photoelectric Instruments Co., Ltd., Yantai 264670, China; 6Shandong Dongyi Institute of Optoelectronic Technology for Industry, Yantai 264670, China

**Keywords:** 3D large-area detector, multiple floating rings, avalanche breakdown, carrier drift paths, charge collection efficiency

## Abstract

The 3D electrode silicon detector eliminates the limit of chip thickness, so it can reduce the electrode spacing (small area) and effectively improve the radiation hardness. In order to expand the application range of the 3D electrode detector, we first propose a 3D large-area silicon detector with a large sensitive volume, and realize multiple floating rings on the upper and lower surfaces of the detector. Due to the influence of different charge states and energy levels in the Si-SiO_2_ interface system, the top and bottom of the 3D P+ electrode are more prone to avalanche breakdown in the 3D large-area detector before the detector is completely depleted or the carrier saturation drift velocity is reached. Moreover, the electric field distribution becomes very uneven under the influence of the oxide charge, resulting in non-equilibrium carriers that cannot drift in the optimal path parallel to the detector surface. In this paper, the effect of floating rings on the performance of a 3D large-area silicon detector is studied by TCAD simulation. It can increase avalanche breakdown voltage by 14 times in a non-irradiated environment, and can work safely in a moderate irradiated environment. The charge collection efficiency can be effectively improved by optimizing the drift path.

## 1. Introduction

A 3D columnar electrode silicon-based detector was first proposed by Professor Parker in 1997 [1]. After more than 20 years of research and development, Centro Nacional de Microelectronica (IMB-CNM, CSIC), the Brookhaven National Laboratory (BNL), Fondazione Bruno Kessler (FBK), and other institutions proposed a range of 3D electrode silicon detector structures and successfully manufactured them by overcoming the deep reactive ion etching technology [2,3,4,5,6]. The 3D electrode silicon detector has excellent radiation resistance because it can be designed with a very small electrode spacing so that carriers can be collected by the collection electrode before being trapped by radiation induced traps. ATLAS inserted a new pixel layer—the Insertable B-Layer (IBL)—during CERN’s first upgrade shutdown for the Large Hadron Collider (LHC), where the radiation fluence reached 1 × 10^16^ n_eq_/cm^2^ (1 MeV neutron equivalent) and the 3D column electrode silicon detector was applied for the first time [7,8,9].

The research of 3D electrode detectors mostly focuses on the structure of a small area (small electrode spacing), which is applied in a high radiation environment. On the contrary, although the 3D large-area detector has lost its strong anti-irradiation performance, increasing the area (large electrode spacing) can effectively reduce the dead zone (low electric field region) caused by deep electrode etching and greatly increase the sensitive volume. Therefore, similar to the excellent performance of large-area silicon drift detector [10,11], the 3D large-area trench detector also has a wide application prospect.

However, when the area of the 3D detector increases, the uniformity of the longitudinal electric field distribution becomes worse due to the influence of the interface charge, and the carrier cannot drift in an ideal path (parallel to the detector surface), which may affect the performance of the detector. In order for the non-equilibrium carriers to drift to the central collecting electrode synchronously, the electric field and potential distribution must be optimized. To make the intrinsic region of the 3D large-area electrode detector fully depleted, a higher reverse bias voltage is needed. The oxide layer charge causes avalanche breakdown before the detector is fully depleted or carriers reach the saturation drift velocity. In addition, if the detector works in a radiation environment, it needs a higher full depletion voltage, so the breakdown voltage of the 3D large-area detector must be increased to ensure that the detector works safely.

In this paper, we propose a 3D large-area trench electrode silicon detector with multiple floating rings (3D-LATD-FRs) based on the 3D trench electrode silicon detector structure proposed by Zheng Li at Brookhaven National Laboratory (BNL) in 2011 [12], as shown in Figure 1. Designing a multiple floating rings construction is an efficient way to address the previous issues. The electrical characteristics, avalanche breakdown voltage, and charge collection of 3D-LATD-FRs will be simulated and analyzed in the following sections.

## 2. Analysis of the Main Problems of 3D Large-Area Detectors

Because the large volume of 3D simulation is too computationally expensive, we use the longitudinal 2D section shown in Figure 1b for subsequent simulations. The 3D detector in Figure 1a can be obtained by rotating the 2D section in Figure 1b by 180°, so the 2D section simulation can effectively reflect the performance of the 3D detector. It can also provide a reference for the 3D detector construction design, such as 3D single-sided cylindrical, 3D double-sided cylindrical, 3D flat-panel detectors, and 3D trench electrode detectors. The thickness of the wafer is 300 μm, the depths of the trench electrode and the central electrode are 270 μm, with the same width of 10 μm. The Bosch process of alternating etching and passivation, which can protect the sidewalls and produce regulated lateral etching, is used to etch the central electrode and the 3D trench electrode [13]. The interface charge density is set at a value of 4 × 10^11^ cm^−2^, the electrode doping concentration is 1 × 10^19^ cm^−3^, and the silicon substrate doping concentration is 1 × 10^12^ cm^−3^. Physical models including the Auger composite model, mobility model, high-field saturation model, band gap narrowing effect, and surface scattering are all available in the simulation. In order to evaluate the avalanche breakdown, the Lackner model [14] is also included in this simulation.

Figure 2 shows the 3D surface diagram of potential distribution and electric field distribution of a 3D large-area detector when the electrode spacing is 500 μm (detector cell area = 0.83 mm^2^, which is 67 times that of a traditional 3D detector with an electrode spacing of 50 μm). The absolute value of reverse bias voltage is 300 V. The structure is NPP type (N-type trench, P-type silicon substrate, P-type central readout electrode) without a floating ring. Influenced by oxide charge, the potential surface in Figure 2a shows a funnel shape, and the potential at the top and bottom of the detector drops rapidly near the central electrode. As shown in Figure 2b, there are high electric fields at both the top and bottom of the central electrode, but the electric field at the bottom is twice that at the top. On the one hand, it is due to the high electric field caused by the tip [15,16,17] at the bottom after deep reaction ion etching; on the other hand, it is due to the influence of the oxide charge at the bottom. The electric field lines from the positive charges in the oxide layer need to terminate at the negative charges in the P-type depletion region. If it is the central P+ electrode, the electric field lines are concentrated in the center, and the local electric field is higher due to the dense electric field lines. When the trench electrode is a P+ electrode, as shown in Figure 2c,d, the total charge in the oxide layer remains unchanged, the electric field lines are divided into two sides, and the highest electric field decreases. Therefore, the breakdown voltage is higher when the trench electrode is P+. It can also be seen in Figure 2e that by increasing the wafer thickness (increasing the distance between the electrode bottom and the oxide layer at the bottom), the high electric field at the bottom of the electrode disappears. However, increasing the detector thickness will also lead to the increase of the detector bulk leakage current. Moreover, CERN requires that the thickness of the 3D detector be as thin as possible (about 100 μm~200 μm), so increasing the detector thickness is not a surefire option to increase the detector avalanche breakdown voltage.

Figure 3 shows the I-V curves of the NPP, NNP, PPN, and PNN structures. The absolute value of the breakdown voltage is less than 400 V when the central electrode is P+, and the absolute value of the breakdown voltage is less than 580 V when the trench electrode is P+. The results correspond to the analysis of the electric field distribution. When designing the device, setting the trench electrode to P+ doping can improve the breakdown voltage, but the effect of the floating ring on the breakdown voltage under the worst condition will be considered in this paper, so the central electrode is set to P+ in the simulation. For the N-type substrate, the surface leakage current increases due to the charge of the oxide layer, and the dark current of the N-type substrate is 1–2 orders of magnitude larger than that of the P-type dark current. At the same time, in order to avoid the inversion of the N-type substrate in the radiation environment, the device is set as a P-type substrate.

## 3. Design and Optimization of Multiple Floating Rings for 3D Large-Area Detector

The floating ring realizes electric field bias through self-division, and its working principle is similar to that of guard ring [18], p-stop [19], FLR [20], and other technologies. The floating rings are P+ concentric rings with the central electrode as the center, made by ion implantation of boron. It mainly achieves: (1) Improve the breakdown voltage of the 3D large-area detector; (2) Optimize the electric field distribution inside the 3D large-area detector to improve the drift path.

### 3.1. Optimization of the Number of Floating Rings and the Ratio of Ring Spacing to Ring Width

The P+ floating rings with different numbers, different widths, and spacings are respectively set on the upper and lower surfaces between the central electrode and the trench electrode of the detector. The floating ring is uniformly doped with a depth of 1 μm, regardless of the process effect. Based on our previous experience, the doping concentration is uniformly set to 1 × 10^17^ cm^−3^.

Firstly, the influence of the number of floating rings on the device performance is studied. The ratio of ring spacing to ring width (*w*:*l*) is fixed at 1:1, the number of floating rings increases from 0 to 12, and the electrical distribution of the detector is shown in Figure 4. We can find that the high electric field at the upper and lower ends of the central electrode disappears after only three floating rings are added, and the positions of avalanche breakdown are assigned to each floating ring, as shown in Figure 4c. As the number of floating rings increases, the electric field and potential distribution become more and more uniform, and 9–12 FRs have no obvious change, as shown in Figure 4a,b. Therefore, if only considering the optimization of electrical characteristics, it is enough to set 12 floating rings. Table 1 shows the breakdown voltage for different numbers of floating rings and different *w*:*l*. When *w*:*l* = 1:1, the breakdown voltage increases with the number of floating rings. The breakdown voltage of 12 floating rings is 845 V, which is twice as high as that of 0 FR. The breakdown voltage should be further improved by increasing the number of floating rings, but the influence of the process parameters of the floating rings has not been considered at this time, and the electrical distribution has reached an ideal state, so the subsequent simulations are set with 12 floating rings.

Next, the influence of different *w*:*l* on the detector performance when the number of 12 floating rings is studied. When *w*:*l* is 3:1, 1:1, 1:3 respectively, the breakdown voltage is 1623 V, 845 V, 272 V, as shown in Table 1. If we only analyze from the simulation results of breakdown voltage, we should choose a larger ratio of ring spacing to ring width. However, from the simulation results in Figure 4, whether it is 3:1 or 1:3, the electric distribution is not as uniform as that of 1:1. Therefore, considering the breakdown voltage and electric distribution, the optimal condition is when the number of floating rings is 12 and *w*:*l* = 1:1. The following research is based on this.

### 3.2. Effect of Floating Ring Process Parameters on Breakdown Voltage in Non-Irradiated Environment

In the actual process, the floating ring is prepared by implanting boron ions with different energies and doses, and the distributions of impurity ions are usually Gaussian. As the reverse bias voltage applied to the N+ trench electrode increases, the inside of the detector is gradually depleted, and the side of the floating ring close to the N+ trench electrode is also partially depleted. Therefore, as shown in Figure 4a, there is a high electric field region on one region of each floating ring, and different process parameters of the floating ring will have a subtle influence on the electric field and thus affect the breakdown voltage.

Other parameters of the detector remain unchanged, and three Gaussian distributions are simulated: (a) Peak position 0.5 μm, junction depth 1 μm; (b) Peak position 0 μm, junction depth 1 μm; (c) Peak position 0 μm, junction depth 0.5 μm. The relationships between peak concentrations of different floating rings and breakdown voltage are shown in Figure 5. The overall trend of (a) and (b) is basically the same, and the highest breakdown voltage (4428 V) when the peak position is on the surface (0 μm) is slightly higher than that (4169 V) when the peak position is 0.5 μm. So by comparing (a) and (b), changing the peak position with the same junction depth has no significant effect on the breakdown voltage. In case (c), when the peak concentration is 7 × 10^16^ cm^−3^, the maximum breakdown voltage increases to 5536 V, and when the junction depth decreases from 1 μm to 0.5 μm, the breakdown voltage increases by 1000 V, which is 14 times higher than the value of 388 V for the detector without a floating ring structure. Therefore, comparing the simulation results of (b) and (c), we find that when the peak positions are all located on the surface, reducing the junction depth and slightly increasing the peak concentration can obtain a higher breakdown voltage. Thence, the influence of process parameters in the actual preparation should also be carefully considered. In the subsequent simulations, the peak position of the floating ring is set on the surface, and the junction depth is 0.5 μm.

Figure 6 shows the electric field distribution curves of the detector surface (thickness = 0 μm) and the electrode bottom (thickness = 270 μm). In Figure 6a, under the same reverse bias voltage of 300 V, compared with the structure without floating rings, the maximum field with floating rings decreases by about 50%. When the reverse bias voltage of 3000 V is applied to the detector with floating rings, the maximum electric field near the top of the central collecting electrode is still smaller than that of a detector without floating rings. As shown in Figure 6b, the floating ring structure has a more obvious alleviating effect on the electric field near the bottom electrode, and the maximum field at 3000 V of a detector with floating rings is about 64% lower than that of a detector without floating rings at 300 V.

### 3.3. Effect of Floating Ring Process Parameters on Breakdown Voltage in Irradiated Environment

The radiation effect causes the detector body damage and surface damage. In this section, the influence of the floating ring on the breakdown voltage of the 3D large-area detector in the radiation environment is studied.

#### 3.3.1. Body Damage

Bulk damage depends on the non-ionizing energy loss (NIEL) of high-energy particles. When the energies of various hadrons or high-energy leptons are higher than the threshold energy of 25 eV in the radiation environment, gaps and vacancy pairs are generated through primary knock-on atoms (PKAs), thus introducing acceptors defect and donors defect in silicon bulk [21,22]. The effective doping concentration(*N_eff_*) of the detector varies with the following equation [23,24,25]:(1)$$N_{eff}\left(\Phi \right)=N_{D0}e^{-C_{D}\Phi }-N_{A0}e^{-C_{A}\Phi }+B_{D}\Phi -B_{A}\Phi$$
where *N_D_*_0_ and *N_A_*_0_ represent the intrinsic doping concentration of the N- and P-type silicon substrate, respectively; *Φ* represents the 1 MeV neutron-equivalent fluence; *C_D_* and *C_A_* represent the shallow donor and shallow acceptor removal rates; and *B_D_* and *B_A_* represent the donor and acceptor deep level defect contribution rates. The depletion voltage is set as minimum working voltage of the detector, which is related to electrode spacing and substrate doping concentration. Substitute Equation (1) into the depletion voltage formula of the 3D trench electrode silicon detector with P-type substrate and P+ doped central electrode:(2)$$V_{fd}\left(\Phi \right)\approx \frac{qb_{A}\Phi }{2\varepsilon_{r}\varepsilon_{0}}\left[\frac{1}{2}\left(R^{2}-R_{c}{}^{2}\right)-R_{c}{}^{2}ln\frac{R}{R_{c}}\right]$$
where *V_fd_* is the depletion voltage; *R* is the detector radius; and *R_c_* is the central collecting electrode radius. When the electrode spacing is constant, *V_fd_* will increase by an order of magnitude as the radiation fluence increases, which is consistent with the experimental results [26]. The detector substrate is ultra-pure high-resistance silicon, and the doping concentration is generally in the order of 10^11^ cm^−3^. After strong radiation, the *N_eff_* of the detector can be increased to 1 × 10^14^ cm^−3^. The breakdown voltage of 300–550 V without the floating ring will not guarantee the long-term stable operation of the 3D large-area detector in the irradiation environment.

Figure 7 shows the graph of the change of breakdown voltage with the peak concentration of floating rings with effective doping concentration *N_eff_* as a parameter varying from 1 × 10^11^ cm^−3^–1 × 10^14^ cm^−3^. When *N_eff_* ≤ 1 × 10^12^ cm^−3^, the three curves are basically coincident. When the peak concentration of the floating rings is in the range of 4–10 × 10^16^ cm^−3^, the breakdown voltage can be kept in the range of 2500–4800 V, and the detector can work in a modest radiation environment with long-term safety. When 1 × 10^12^ cm^−3^ < *N_eff_* < 1 × 10^13^ cm^−3^, the breakdown voltage will change significantly with peak concentration. The maximum breakdown voltage is at the peak concentration of the floating rings at 6 × 10^16^ cm^−3^. However, the effect of floating rings on improving breakdown voltage decreases in a high radiation environment. For *N_eff_* = 5 × 10^13^ cm^−3^–1 × 10^14^ cm^−3^, the floating rings only slightly increase the detector breakdown voltage. Therefore, the 3D large-area detector can work in the irradiation environment with radiation fluence *Φ* ≤ 1 × 10^15^ n_eq_ cm^−2^.

#### 3.3.2. Surface Damage

In the CMOS process, SiO_2_ is grown by dry oxidation or wet oxidation:Dry: Si + O_2_ → SiO_2_
Wet: Si + 2H_2_O → SiO_2_ + 2H_2_

The SiO_2_ grown by dry oxidation presents a glassy shape, showing a short-range order, and the quality is usually better than that of wet oxidation. In the current process, the oxide charge density can be controlled in the order of 10^11^ cm^−3^. In the irradiation environment, the saturated oxide charge density is up to 3 × 10^12^ cm^−2^ [27]. In the high-dose gamma irradiation experiment, due to the combination of interface damage and bulk damage, the saturation oxide charge density does not increase as expected when it reaches 1 × 10^12^ cm^−2^ [28], but even a small increase of the oxide charge density will have a great impact on the breakdown voltage and electric field distribution.

To improve the charge collection efficiency (CCE) of 3D large-area electrode silicon detectors, defects caused by radiation or high-energy particles in silicon should be reduced or avoided. The threshold energy of X-rays for bulk damage to silicon is about 300 KeV [29,30]. Therefore, high-dose X-rays mainly causes surface damage to the detector, and the 3D large-area electrode silicon detector is more suitable for hard X-ray detection. Every 18 eV of high-energy X-ray passing through the SiO_2_ generates an electron-hole pair within the oxide layer. Electrons drift rapidly out of the oxide layer driven by the electric field, while holes have extremely low mobility in silicon dioxide and can only diffuse to the Si-SiO_2_ interface, resulting in an increase in the total positive oxide charge. High doses of X-rays produce saturated oxide charge density up to about 2 × 10^12^ cm^−2^ [31] in oxide.

Figure 8 shows the breakdown voltage curves corresponding to different peak concentrations of the floating rings when the oxide layer thickness is 1 μm, and the oxide charge density increases from 4 × 10^11^ cm^−2^ to 2 × 10^12^ cm^−2^. As the oxide charge density increases, the improvement of the breakdown voltage by the floating ring gradually decreases. The optimal peak concentration of the floating ring corresponding to the highest breakdown voltage increases with the increase of the oxide charge density, that is, a higher negative charge is required to neutralize the positive charge density in the oxide layer. Therefore, a higher peak concentration should be selected when a large-area 3D detector needs to work in an irradiated environment. For example, when the peak concentration is 1.4 × 10^17^ cm^−3^, the breakdown voltage is no longer the highest, but it can still remain around 2000 V under moderate radiation fluence.

### 3.4. Comparison of Charge Collection before and after Drift Path Optimization

Response to the minimum ionizing particles (MIP) is an important parameter to measure the detector performance. The energy band gap of silicon is 1.12 eV. Since silicon is an indirect band-gap semiconductor, the electron-hole pair generation process requires phonons participation, so the average energy of electron-hole pair generation in silicon is 3.6 eV, which is three times the bang gap. A MIP produces an average of 80 electron-hole pairs per micron in its incident trajectory in the detector.

As shown in Figure 9, when high-energy particles or photons are incident perpendicular to the detector surface, electron-hole pairs are created in the detector bulk along a trajectory parallel to the detector electrodes. For the P+ collecting electrode in our case, the readout current is composed of induced currents caused by drifting of holes and electrons in the detector. According to the non-equilibrium continuity equation:(3)$$\frac{\partial \Delta p}{\partial t}=D_{P}\frac{\partial^{2}\Delta p}{\partial x^{2}}-\mu_{P}\vec{E}\left(x\right)\frac{\partial \Delta p}{\partial x}-\mu_{P}p\frac{\partial \vec{E}\left(x\right)}{\partial x}-\frac{\Delta p}{\tau_{P}}$$

Non-equilibrium carriers drift toward the central collector electrode driven by the electric field. According to the Shockley–Ramo theorem [32], the induced current is related to the drift velocity. Considering the saturation drift velocity, the hole drift velocity [33] in the detector is:(4)$$v_{dR}^{e,h}=\frac{dR^{e,h}}{dt}=\frac{\mu^{e,h}E\left(R^{e,h}\right)}{1+\frac{\mu^{e,h}E\left(R^{e,h}\right)}{v_{s}}}$$

The induced current is affected by the distribution and magnitude of the electric field of the detector. Here, we divide the detector into three parts: top, middle, and bottom parts according to the schematic diagram in Figure 9. The figure shows four incident trajectories, two of which are incident at *D* = 250 μm and *D* = 500 μm for α = 90°. The other two are incident at the positions of *D* = 250 μm and *D* = 500 μm for α = 45° and α = 135°, respectively, and show a symmetrical distribution. The total number of electron-hole pairs produced at 45° or 135° incidence is greater than that at normal incidence.

Figure 10 and Figure 11 present the distributions of hole concentration and hole currents at different times. Figure 10 is the dynamic distribution diagram of hole concentration and hole current density without the 12 floating rings. For Figure 12a,b, at the initial moment, MIP penetrates vertically at *D* = 250 μm and generates electron-hole pairs in the incident trajectory. The middle part of the detector (near thickness = 150 μm) is less affected by the Si-SiO_2_ interface system and can be used as the optimal drift path. Due to the lower electric field in the top and bottom of the detector and the higher electric field in the middle, the drift velocity of the non-equilibrium holes in the top and bottom of the detector is smaller than that in the middle, and the holes cannot drift to the central electrode synchronously. Therefore, some holes are trapped by interface states or deep levels in the body, and some of them may even drift toward the middle of the detector instead of the collection electrode. The longer drift path also leads to a higher hole trapping rate, which ultimately leads to a lower CCE. In Figure 10c,d, the incident is perforated at an angle *α* = 45° at *D* = 250 μm. As a whole, a narrow drift channel is formed in the middle of the detector. It can be found that at the end of the incident trajectory, the hole basically does not drift toward the center because the electric field at the bottom of the detector is very low. Figure 11 is the dynamic distribution diagram of hole concentration and hole current density after adding the 12 floating rings. The holes generated in the incident trajectory drift to the central electrode synchronously, and the holes on the upper and lower surfaces are not recombined or trapped in large quantities. Good consistency is maintained even at incidence at angle α = 45°.

In the following part of the study we will mainly study the influence of drift path on the CCE, the influence of deep level capture center [28,34] on the CCE is ignored in the simulation for clarity. For a small area 3D trench electrode silicon detector, the CCE is independent of incident position in a non-radiation environment [33], but the CCE of a 3D large-area trench electrode silicon detector will be very different in different incident positions, as shown in Figure 12a. To ensure the safe operation of a 3D large-area detector without the floating ring, a 300 V reverse bias voltage is applied. The CCEs of incidence of MIP at *D =* 250 μm and *D* = 500 μm are 70% and 45%, respectively. After adding the floating ring to the detector, the CCE increases by about 5% at the same reverse bias voltage. The reverse bias voltage of 3D-LATD-FRs can be further increased. For *D* = 250 μm incidence, the CCE can reach 130% at 1500 V due to the avalanche effect. For *D* = 500 μm incidence, the CCE reached 80% at 1500 V, which is 35% higher than that without the floating ring structure at the same position. In Figure 12b, when the bias voltage is 300 V, the CCE is increased by 15–20% after adding the floating ring, and the optimization of the drift path greatly improves the CCE. At the same time, it is found that the CCE at *D* = 250, α = 45° incidence is lower than that at *D* = 500, α = 135° incidence, this is because the end of the incident path enters the outside of the detector when α = 45°, resulting in part of the signal loss, which can be seen in Figure 10 and Figure 11c,d.

## 4. Conclusions

In this work, the breakdown voltage and optimal drift path of a 3D large-area trench electrode silicon detector are studied by numerical simulations. For the 3D large-area detector without a floating ring structure, avalanche breakdown occurs easily at the upper and lower ends of the P+ collection electrode, and the non-equilibrium carriers cannot drift in the optimal path, thus the CCE decreases.

By comparison, it is found that when the ratio of ring spacing to ring width (*w*:*l*) is 1:1, the more the number of floating rings, the more uniform the electric field and potential distribution, and the higher the breakdown voltage. With the increase of the ratio of ring distance to ring width, the breakdown voltage will be further increased, but the optimal distribution of electric field and potential will be affected, and the optimal situation between them may need to be further refined. The multiple floating rings are implanted with boron ions and usually exhibit a Gaussian distribution. Through simulation in a non-radiation environment, it is found that the closer the peak position is to the surface and the shallower the junction depth, the higher the avalanche breakdown voltage. When the charge density of the oxide layer is 4 × 10^11^ cm^−2^, the peak concentration is the most effective for the improvement of the breakdown voltage when the peak concentration is in the order of 16 times, and the maximum can be increased by 14 times. Through the simulation in the radiation environment, it is found that the radiation hardness of the 3D large-area detector is improved after the floating ring structure is applied, and it can still work safely in the medium radiation environment. After the electric field and potential distribution of the detector are optimized, the holes generated in the upper, middle, and bottom parts drift to the central collecting electrode synchronously. In simulations, it is found that the CCE can be significantly improved just by optimizing the drift path.

## Figures and Tables

**Figure 1 sensors-22-06352-f001:**
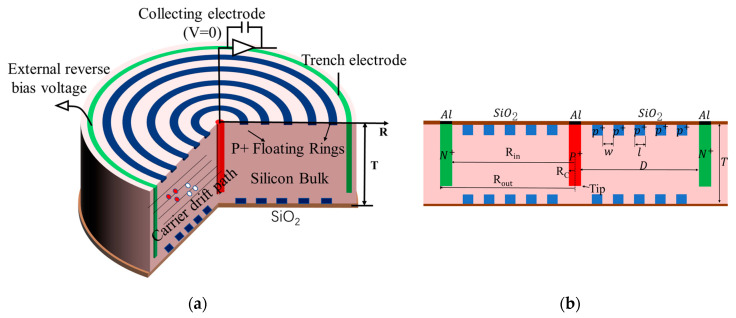
Structure diagram: (**a**) 3D large-area trench electrode silicon detector with multiple floating rings; and (**b**) longitudinal cross-section.

**Figure 2 sensors-22-06352-f002:**
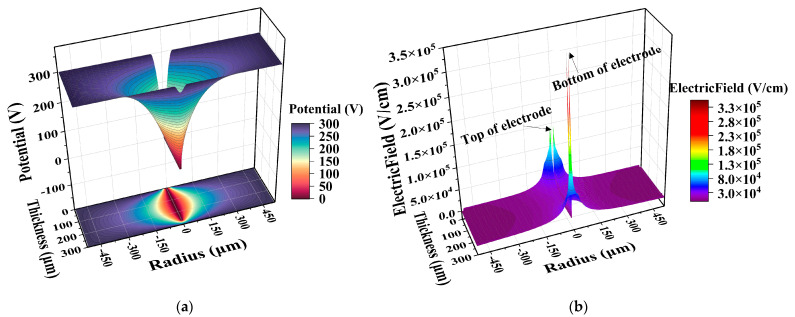
The three-dimensional surface diagram: (**a**) NPP-type potential distribution; (**b**) NPP-type electric field distribution; (**c**) PNN-type potential distribution; (**d**) PNN-type electric field distribution; (**e**) Electric field distribution of wafer thickness 750 μm.

**Figure 3 sensors-22-06352-f003:**
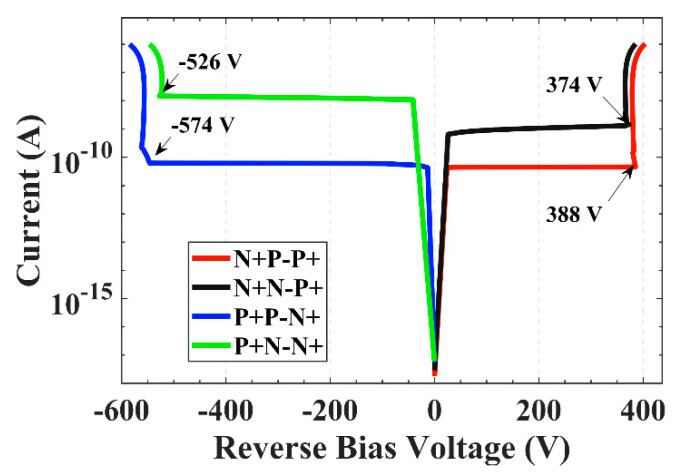
*I–V* curves of four doping types.

**Figure 4 sensors-22-06352-f004:**
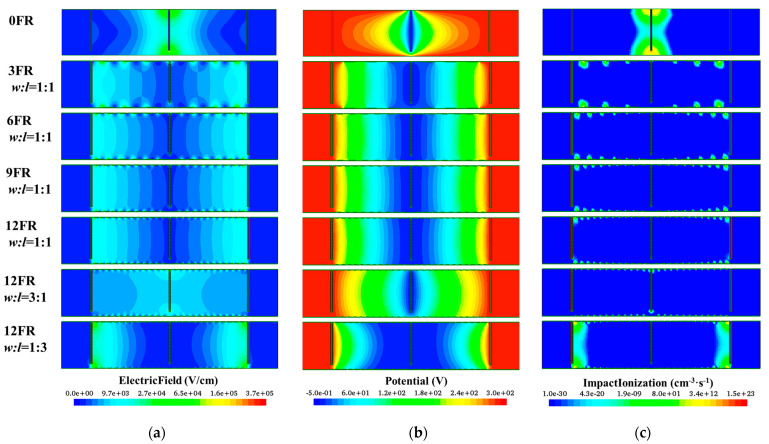
The 3D large-area detectors with different numbers of floating rings and different ratios of ring width and ring spacing with an applied reverse bias voltage of 300 V: (**a**) Electric field distribution; (**b**) Potential distribution; (**c**) Impact ionization distribution.

**Figure 5 sensors-22-06352-f005:**
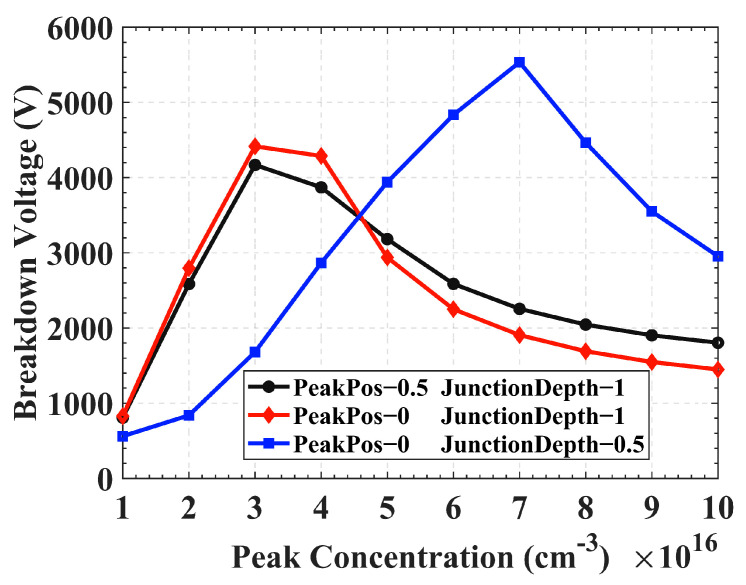
Influence of floating ring process parameters on breakdown voltage.

**Figure 6 sensors-22-06352-f006:**
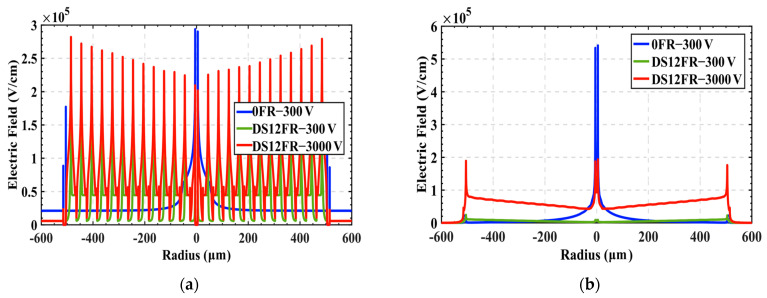
Electric field distribution curves: (**a**) detector surface (thickness = 0 μm); (**b**) electrode bottom (thickness = 270 μm).

**Figure 7 sensors-22-06352-f007:**
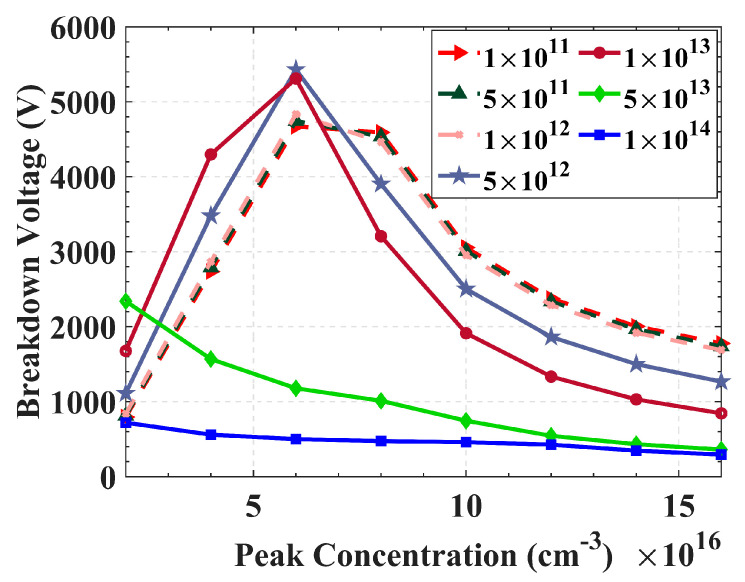
The relationship between peak concentration of floating rings and breakdown voltage at different effective doping concentrations (*N_eff_*).

**Figure 8 sensors-22-06352-f008:**
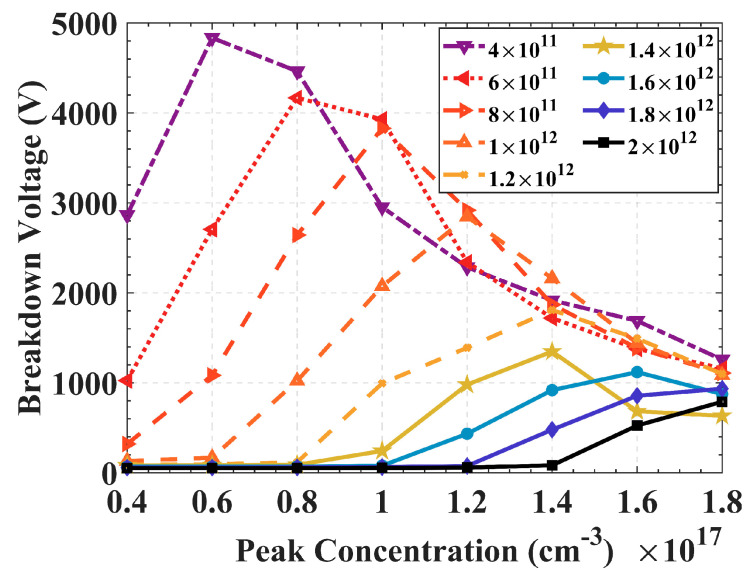
Relationship between peak concentration of floating ring and breakdown voltage at different oxide charge densities.

**Figure 9 sensors-22-06352-f009:**
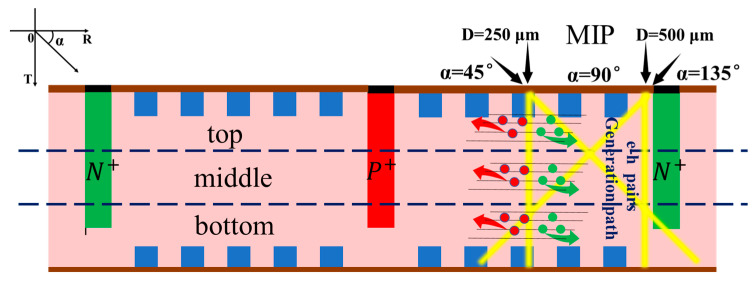
Schematic diagram of the incidence of MIP at different positions.

**Figure 10 sensors-22-06352-f010:**
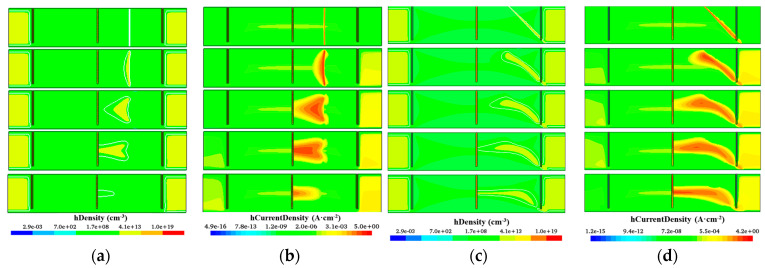
Hole density and hole current density at different times of a 3D large-area trench electrode silicon detector without floating ring: (**a**) Hole density, α = 90°; (**b**) Hole current density, α = 90°; (**c**) Hole density, α = 45°; (**d**) Hole current density, α = 45°.

**Figure 11 sensors-22-06352-f011:**
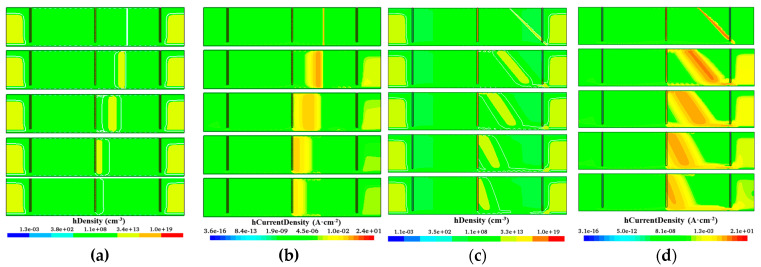
Hole density and hole current density at different times of a 3D large-area trench electrode silicon detector with floating ring (**a**) Hole density, α = 90°; (**b**) Hole current density, α = 90°; (**c**) Hole density, α = 45°; (**d**) Hole current density, α = 45°.

**Figure 12 sensors-22-06352-f012:**
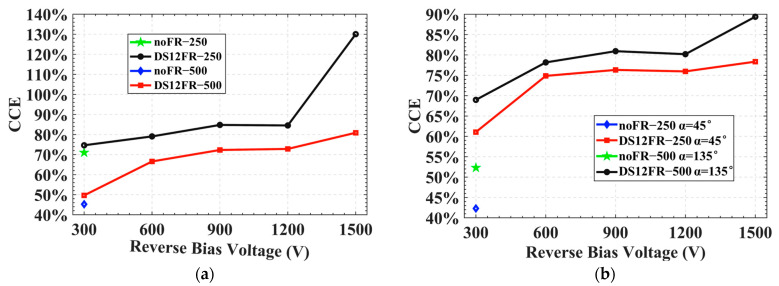
Charge collection efficiency: (**a**) Incident at *D* = 250 μm and *D* = 500 μm vertical surfaces, respectively; (**b**) Incident at α = 45° and α = 135°, respectively.

**Table 1 sensors-22-06352-t001:** Device simulation results of breakdown voltage with different numbers and ratios of floating rings in N+P−P+ structure.

FR	*w*:*l*	Breakdown Voltage (V)
0 FR	-	388
3 FR	1:1	483
6 FR	1:1	671
9 FR	1:1	770
12 FR	1:1	845
	3:1	1623
	1:3	272

## Data Availability

The data presented in this study are available on request from the author.

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
