# Peer review of "Optimal Design of Multiple Floating Rings for 3D Large-Area Trench Electrode Silicon Detector"

_sensors, 2022, doi:10.3390/s22176352_

Round 1

Reviewer 1 Report

In order to improve the radiation tolerance, breakdown voltage and CCE of 3D large-area trench electrode silicon detector, the paper proposed the optimal design of multiple floating rings. After adding the floating rings, some performance parameters are greatly improved, which can be drawn from the simulation results. However, some explanations about the optimized ways are lacked. Some writing errors are also founded.

Detailed comments:

1. Abstract: anti-radiation -> radiation hardness or radiation tolerance

2. Introduction: Other published works about the large area 3D detector are suggested to be mentioned here, which can prove this work more meaningful.

3. L75: It seems to be that longitudinal 2D section replace 3D section in the simulation. Will the simulation accuracy be decreased?

4. L87: This sentence maybe clearer if modified to: In order to evaluate the avalanche breakdown, the Lackner model [12] is also included in this simulation.

5. L104-106: This sentence is not very clear to explain the results shown in Fig.2(b).

6. L110: It seems that the detector is designed for CERN. The design specification of detector is suggested to be added, if the application is determinate.

7. L113: In the NNP, NPP, PPN and PNN structures ...

8. Section2: The section title is the problem of increasing area. It is better to give the value of area, which is considered to be large. The content just gives some results. The problems about breakdown voltage and CCE maybe not given.

9. L117: Could you please explain the reason why the breakdown voltage is higher when the trench electrode is P+?

10. L135: The P+ floating rings are chosen in the optimization. Is it concluded from the results shown in Fig.3? More expressions are suggested to give more information about the detector design.

11. L145: Three Gaussian distributions were simulated. Why the values in these cases are chosen? The breakdown voltage is improved by the floating rings according to Fig.4. The discussion and explanation are suggested to be given.

12. L184: Figure 6 shows ...

13. L211:CD ->CD, CA->CA

14. L356-357: CCE is increased by about 5% after adding the floating ring and it can be improved by increasing bias voltage. Are there other methods to improve the CCE. For example, the number or the space of the floating rings. It is suggested to give more information about the principle of CCE improved by floating rings.

15. This title of this paper is the optimal design of 3D-LATD-FRs. However, many simulation results are discussed by comparing the difference between with and without floating rings. More information of the optimization designs about the 3D large-area trench are hoped to be given in the section 3. The main optimized factors are suggested to be clearly introduced.

Reviewer 2 Report

[General comments]

This is important work that can offer advice on theory for actual applications. Software simulation allows for the optimized design of multiple floating rings for three-dimensional large-area trench electrode silicon detectors. However, there are some problems with the structure, content and language in this paper. It is suggested that the authors should make the corresponding revisions after reading the suggestions.

[Major comments]

1.      In Section 3.1.2, line 278-281.This statement is overly long and difficult to read. Additionally, it is difficult to grasp how the phrase - "giving long enough detector working time"- connects to the context because it appears too abruptly. Therefore I recommend removing it.

2.      In Section 3. It is recommended that the title of part 3.1.2 corresponds to the title of part 3.1.1. Although the title of part 3.1.2 emphasizes the effect of body damage and surface damage on breakdown voltage, this section actually focuses on the impact of radiation on breakdown voltage. It would be more accurate to change the title to "Effect of floating ring process parameters on breakdown voltage in irradiated environment". In addition, after reading the article as a whole, the author should adjust the structure of the third part to avoid unbalanced structure of the article.

3.      Page 2, line 76. Is the simulation of the 3D detector obtained from the 2D section by rotation? If so, it is recommended to indicate this in the article.

4.      Page 2, line 353. CCE has different results at different incidence positions, please explain why it is not required to take the angle of incidence into account.

[More detailed comments]

1.      Page 7, line 209. Please explain the meaning of Neff when it first appears, so that readers will comprehend it. For CEE below, the same is valid.

2.      Page 7, lines 235-239. The peak concentration of the floating rings do not correspond to the value in Figure 7, please revise them carefully.

3.      The name of the legend located below in Figure 10 do not match that in Figure 11, so please take care to ensure consistency of comparison.

4.      Please carefully adjust the image content's format to comply with the magazine's formatting requirements.

Round 2

Reviewer 2 Report

The manuscript has been revised according to the comments, I agree to accept this manuscript of current version.